# Astrocytes contribute to synapse elimination *via* type 2 inositol 1,4,5-trisphosphate receptor-dependent release of ATP

**Junhua Yang[†], Hongbin Yang[†], Yali Liu, Xia Li, Liming Qin, Huifang Lou, Shumin Duan\*, Hao Wang\***

Department of Neurobiology, Key Laboratory of Medical Neurobiology of Ministry of Health of China, Key Laboratory of Neurobiology, Zhejiang University School of Medicine, Hangzhou, China

**Abstract** Selective elimination of unwanted synapses is vital for the precise formation of neuronal circuits during development, but the underlying mechanisms remain unclear. Using inositol 1,4,5-trisphosphate receptor type 2 knockout ($Itpr2^{-/-}$) mice to specifically disturb somatic $Ca^{2+}$ signaling in astrocytes, we showed that developmental elimination of the ventral posteromedial nucleus relay synapse was impaired. Interestingly, intracerebroventricular injection of ATP, but not adenosine, rescued the deficit in synapse elimination in $Itpr2^{-/-}$ mice. Further studies showed that developmental synapse elimination was also impaired in $P2ry1^{-/-}$ mice and was not rescued by ATP, indicating a possible role of purinergic signaling. This hypothesis was confirmed by MRS-2365, a selective P2Y1 agonist, could also rescue the deficient of synapse elimination in $Itpr2^{-/-}$ mice. Our results uncovered a novel mechanism suggesting that astrocytes release ATP in an IP3R2-dependent manner to regulate synapse elimination.

**\*For correspondence:**
duanshumin@zju.edu.cn (SD);
haowang@zju.edu.cn (HW)

[†]These authors contributed equally to this work

**Competing interests:** The authors declare that no competing interests exist.

## Introduction

Synapse elimination, a process of pruning inappropriate synapses during development is essential for the formation of neuronal circuits and proper brain function (*Katz and Shatz, 1996*; *Paolicelli et al., 2011*; *Sanes and Lichtman, 1999*). Disruption of this process is likely involved in many neurological diseases such as schizophrenia and autism (*Hoffman and McGlashan, 1997*; *Tang et al., 2014*; *Tsai et al., 2012*). Although activity-dependent competition among different inputs has been suggested as the key mechanisms for synapse elimination, the underlying molecular and cellular mechanisms remain unclear. In addition to the retino-geniculate and climbing fiber-Purkinje cell pathways (*Chen and Regehr, 2000*; *Kano and Hashimoto, 2009*), the principal trigeminal nucleus-ventral posteriormedial thalamic nucleus (Pr5-VPm) connection in mice has proven to be an excellent model in which to investigate developmental synapse elimination in the central nervous system (*Arsenault and Zhang, 2006*). VPm neurons receive a contra-lateral Pr5 projection and relay the information to the somatosensory cortex. At an early age like postnatal day 7 (P7), each VPm neuron receives an average of 7–8 Pr5 inputs. Most of these inputs are eliminated in an activity-dependent manner during development and the majority of VPm neurons only receive a single Pr5 input by P16.

Over the past decade, emerging evidence has suggested that astrocytes not only provide structural and metabolic support to neighboring neurons, but also participate in synapse formation, synaptic transmission, and synaptic plasticity (*Allen, 2014*; *Henneberger et al., 2010*; *Zhang et al.,*

**eLife digest** Neighbouring neurons connect to each other and share information through structures known as synapses. As the brain develops, many synapses turn out to be redundant. Just like trees in a garden that need to be trimmed, these redundant synapses must be pruned in order to form the right pattern of connections between different neurons. Brain cells called astrocytes play a key role in synaptic pruning, but it is unclear exactly how astrocytes coordinate this process.

One important way in which astrocytes communicate with neurons is through a process called calcium signaling, in which the movement of calcium ions into or out of the cell sets off a cascade of activity inside the astrocytes. Yang et al. have now studied developing mice that lacked a gene that is essential for calcium signaling in astrocytes. Two weeks after they were born, these mice still had redundant synapses that are normally lost after birth. However, injecting the developing brain with a substance called ATP prevented this defect and allowed synapses to be correctly pruned. This is likely to be because astrocytes also use ATP to communicate with neurons, and ATP compensated for the missing calcium signaling.

The experiments also uncovered the specific structure – called the P2Y1 receptor – on the outer surface of a neuron that ATP latches on to in order to help remove synapses. Further work is now needed to reveal how activating the P2Y1 receptor coordinates synaptic removal.

*2003*). Structurally, astrocytes contact 50–90% of the synapses in a given brain region (*Genoud et al., 2006*; *Oliet et al., 2001*) and form one component of the "tripartite synapse" (*Araque et al., 1999*). Functionally, intracellular $Ca^{2+}$ ($[Ca^{2+}]_i$) elevation in neighboring astrocytes can be induced by activation of the synapse, triggering the release of gliotransmitters such as ATP and D-serine that modulate synaptic efficacy and plasticity (*Chen et al., 2013*; *Fields and Burnstock, 2006*; *Henneberger et al., 2010*; *Yang et al., 2003*). It is generally accepted that astrocytes release gliotransmitters in a G-protein-coupled receptor/ inositol 1,4,5-trisphosphate receptor (GPCR/IP3R)-mediated $Ca^{2+}$-dependent manner. Studies in both acute slice preparations and in vivo have shown that, in response to neuronal activity, GPCR-mediated $Ca^{2+}$ signaling in astrocytes depends on the activation of IP3R2, a subunit that is solely expressed in astrocytes (*Fiacco and McCarthy, 2004*; *Navarrete et al., 2012*). Controversially, studies also indicated that IP3R2-mediated $Ca^{2+}$ signaling in astrocytes may not be involved in the acute modulation of neuronal activity and synaptic transmission (*Agulhon et al., 2010*; *Petravicz et al., 2008*). Indeed, there are other $Ca^{2+}$ signaling pathways in astrocytes which do not involve IP3R2. A recent study showed that although the somatic $Ca^{2+}$ signaling was eliminated, $Ca^{2+}$ signaling in processes was intact in the astrocyes of $Itpr2^{-/-}$ mice (*Srinivasan et al., 2015*). Therefore, IP3-induced $Ca^{2+}$ increase is only the major source for somatic $Ca^{2+}$ signaling in astrocytes. Recent studies have shown that astrocytes could mediate synapse elimination in a neuronal activity-dependent manner through two distinct pathways: either directly by phagocytosis through the MEGF10 and MERTK pathways, or by activating the classical complement cascade *via* astrocyte-derived transforming growth factor-β (TGF-β) (*Bialas and Stevens, 2013*; *Chung et al., 2013*). Here, we hypothesized that IP3R2-dependent $Ca^{2+}$ signaling in astrocytes might be involved in regulating developmental synapse elimination. The results from studies in the cerebellum suggested that $Ca^{2+}$-permeableα-amino-3-hydroxy-5-methyl-4-isoxazolepropionic acid receptors (AMPARs) in Bergmann glia are required for the developmental elimination of climbing fiber-Purkinje cell inputs, which provides a clue that astrocytic $Ca^{2+}$ signaling might be involved in synapse elimination (*Iino et al., 2001*). Unfortunately, it is difficult to draw an exclusive conclusion since Bergmann glia are morphologically distinct from astrocytes and detect synaptic activity at cerebellar synapses primarily though activation of $Ca^{2+}$-permeable AMPARs as well as GPCRs.

By combining electrophysiological, pharmacological, and immunohistochemical methods, we discovered that selectively disturbing $[Ca^{2+}]_i$ signaling in astrocytes using $Itpr2^{-/-}$ mice impaired the developmental elimination of VPm relay synapses. Intracerebroventricular injection of ATP, but not adenosine, rescued the impaired synapse elimination in these mice. We further found that developmental synapse elimination was also impaired in $P2ry1^{-/-}$ mice and could not be rescued by ATP. Last, the deficit of synapse elimination in $Itpr2^{-/-}$ mice was rescued by intracerebroventricular

injection a selective P2Y1 receptors agonist MRS-2365. Overall, we provide direct evidence to show that astrocytes contribute to synapse elimination in an IP3R2-dependent manner through activation of purinergic signaling.

## Results

### Synapse elimination is disrupted at the VPm relay synapse in *Itpr2$^{-/-}$* mice at P16-17

To determine the role of astrocytic Ca$^{2+}$ signaling in synapse elimination, we need an effective way to selectively disrupt the function of astrocytes. In mammals, astrocytes rely on IP3R2-mediated intracellular Ca$^{2+}$ signaling to perform their functions. A large literature suggested that IP3R2 is the only subtype astrocytes expressed and is not expressed in microglia and neurons in the cerebrum. Thus *Itpr2$^{-/-}$* mice could be used to study the specific roles of Ca$^{2+}$ signaling in astrocytes for synapse function (*Hertle and Yeckel, 2007*; *Li et al., 2015*; *Sharp et al., 1999*). In line with previous studies, we also found that IP3R2 was co-expressed with GFAP, but not bio-markers for microglia or neurons in the brain areas we tested including hippocampus (*Figure 1—figure supplement 1*). Using Ca$^{2+}$ imaging in acute brain slices, we next found that the ATP -induced somatic [Ca$^{2+}$]$_i$ elevation in the astrocytes but not in the neurons of *Itpr2$^{-/-}$* mice (*Figure 1—figure supplement 2b,c*) was abolished in both of the VPm and hippocampus, confirming that Ca$^{2+}$ signaling was selectively impaired in astrocytes in *Itpr2$^{-/-}$* mice. Next, we examined developmental synapse elimination by whole-cell patch recording in acute brain slices. Interestingly, we found a marked difference in the mean number of inputs received by each VPm neuron between WT and *Itpr2$^{-/-}$* mice at P16-18 (WT = 1.2 ± 0.02, n = 26 cells from 4 mice; *Itpr2$^{-/-}$* = 2.1 ± 0.10, n = 40 cells from 6 mice; p<0.01, *Figure 1d*). In WT mice, only 27% (7 of 26) of VPm relay neurons received multiple Pr5 inputs at this age (*Figure 1a,c*), whereas most of these neurons (72%, 32 of 42) in *Itpr2$^{-/-}$* mice received multiple Pr5 inputs (*Figure 1b,c*). VPm relay neurons receive two major excitatory inputs: from layer VI cortex and the other from the Pr5 that express vesicular glutamate transporter 1 (VGluT1) and VGluT2, respectively (*Graziano et al., 2008*). Each Pr5 input forms multiple synaptic contacts with VPm neurons and thus, the number of inputs indicates how many axonal projections while the VGluT2 staining represents number of synaptic terminals. The pruning of somatic innervations by Pr5 inputs in the VPm is always related to the elimination of VPm relay synapses, as showed by previous studies (*Takeuchi et al., 2014*; *Zhang et al., 2012*). To further verify that there were more synapses in KO mice, we immunostained for VGluT2. Consistent with the electrophysiological results, we observed more VGluT2 puncta around the soma as well as the total numbers of puncta in *Itpr2$^{-/-}$* mice (*Figure 1e,f*), indicating that there were more synapses in KO mice. In addition, we found that neuron number did not significantly change in the VPm between WT and *Itpr2$^{-/-}$* mice at this age (*Figure 1—figure supplement 3*). These results strongly suggested that astrocytic IP3R2-dependent Ca$^{2+}$ signaling is required for synapse elimination.

During development, the strengthening of immature synapses often concurrent with the removal of redundant inputs. In this study, we did not find any change in maximal AMPAR-EPSCs, N-methyl-D-aspartate receptor (NMDA-) EPSCs and APMAR-EPSC/NMDAR-EPSC ratios recorded at VPm relay synapses when WT mice were compared with *Itpr2$^{-/-}$* at P16 (*Figure 1—figure supplement 4*). Given that the *Itpr2$^{-/-}$* mice had more inputs and synapses, the strength of each input or individual synapse was weaker than that in WT mice (*Figure 1—figure supplement 4b*). These data revealed that the deficit of synapse elimination was sufficient to affect synaptic strengthening during development.

### Connectivity of Pr5-VPm pathway is not affected in *Itpr2$^{-/-}$* mice at P7

To distinguish whether the deficit in synapse elimination was due to the failure of developmental elimination or abnormal synaptogenesis at an early age, we tested the connectivity of the Pr5-VPm projection at P7. We found that the number of inputs received by VPm neurons was comparable in the WT and *Itpr2$^{-/-}$* mice (*Figure 2a–d*). Neither the numbers of VGluT2 puncta per soma nor per neuron differed between WT and *Itpr2$^{-/-}$* mice at this age, indicating similar numbers of synapses in these mice (*Figure 2e,f*). These results suggested that synapse formation was normal at early postnatal stage in *Itpr2$^{-/-}$* mice. To address the possibility that the deficit in synapse elimination we

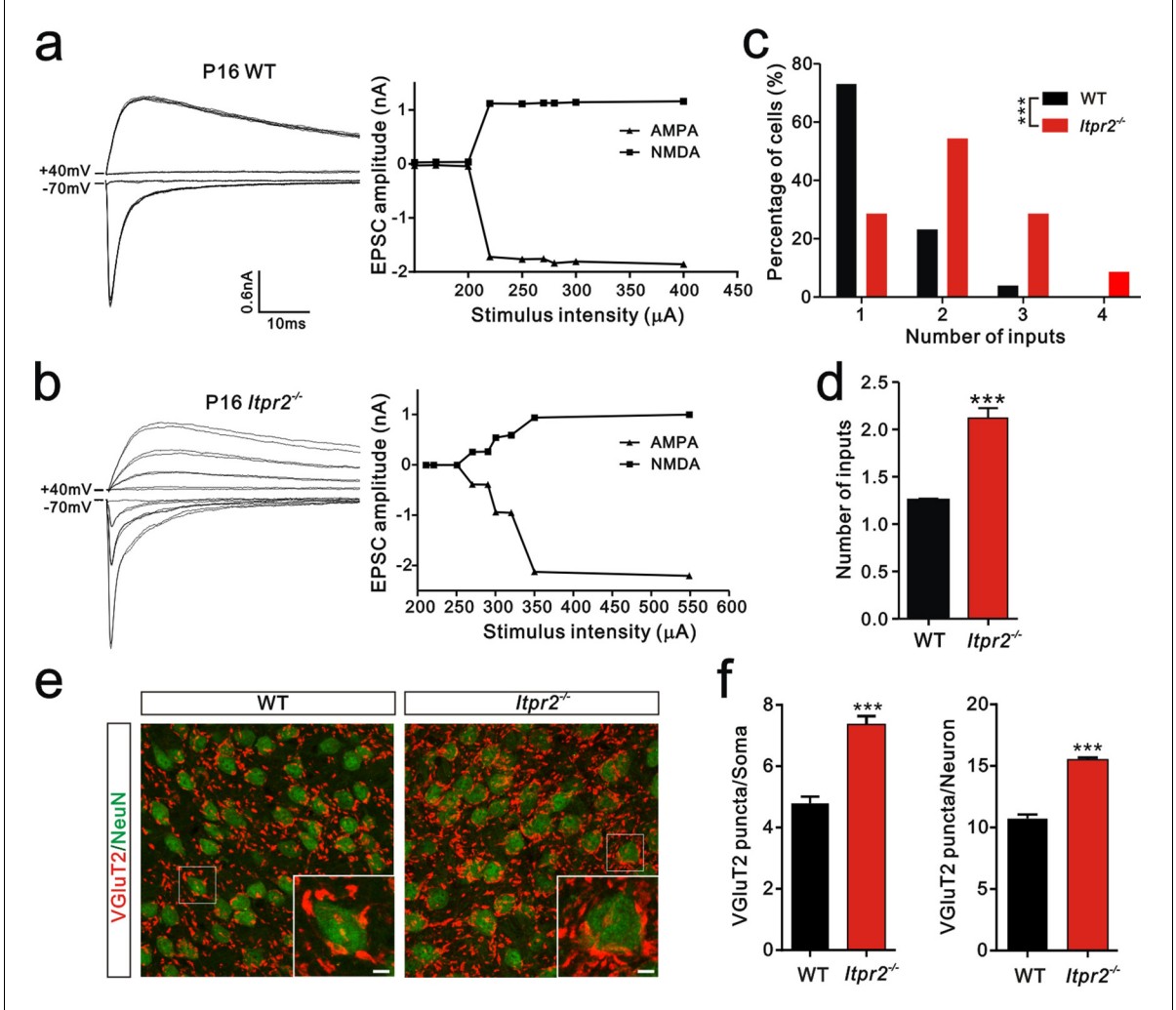

**Figure 1.** Developmental synapse elimination was impaired in *Itpr2*$^{-/-}$ mice at P16-17. (**a** and **b**) Left panels, sample traces of membrane current in response to stimuli over a range of intensities in VPm neurons at P16-17 in WT (**a**) and *Itpr2*$^{-/-}$ mice (**b**). Currents recorded at +40 mV are mediated by NMDA receptors, and those at -70 mV are mediated by AMPA receptors. Right panels, peak current *versus* stimulus intensity for WT (**a**) and *Itpr2*$^{-/-}$ mice (**b**). (**c**) Distributions of the number of Pr5 axons innervating each VPm neuron during P16-17 in WT (n = 26 cells) and *Itpr2*$^{-/-}$ (n = 42 cells) mice. ***p<0.001, $\chi^2$ test. (**d**) Histogram of average number of inputs received by each VPm neuron at P16-17 in WT and *Itpr2*$^{-/-}$ mice. ***p<0.001, unpaired Student's t test. Error bars indicate SEM. (**e**) Sample confocal images of immunostained neurons and Pr5 axonal terminals in the VPm at P16. Neurons were labeled by the NeuN antibody (green), and Pr5 axon terminals were labeled by the VGluT2 antibody (red). Inset is higher-magnification of the boxed area. Scale bar, 5 μm. (**f**) Quantification of VGluT2 puncta/soma (left, n = 40 cells/group) and VGluT2 puncta/neuron (right, n = 12 sections from 4 mice/group) for WT and *Itpr2*$^{-/-}$ mice. ***p<0.001, unpaired Student's t test. Error bars indicate SEM.

The following figure supplements are available for figure 1:

**Figure supplement 1.** Confocal images showing that IP3R2 was specifically expressed in GFAP-positive astrocytes but not GFP-positive microglia.

**Figure supplement 2.** Knocking out IP3R2 specifically disturbed [Ca$^{2+}$]$_i$ elevation in astrocytes in both of hippocampus and the VPm.

**Figure supplement 3.** Neuron number does not change in the VPm between WT and *Itpr2*$^{-/-}$ mice.

**Figure supplement 4.** Synaptic properties are not altered in *Itpr2*$^{-/-}$ mice at P16-17.

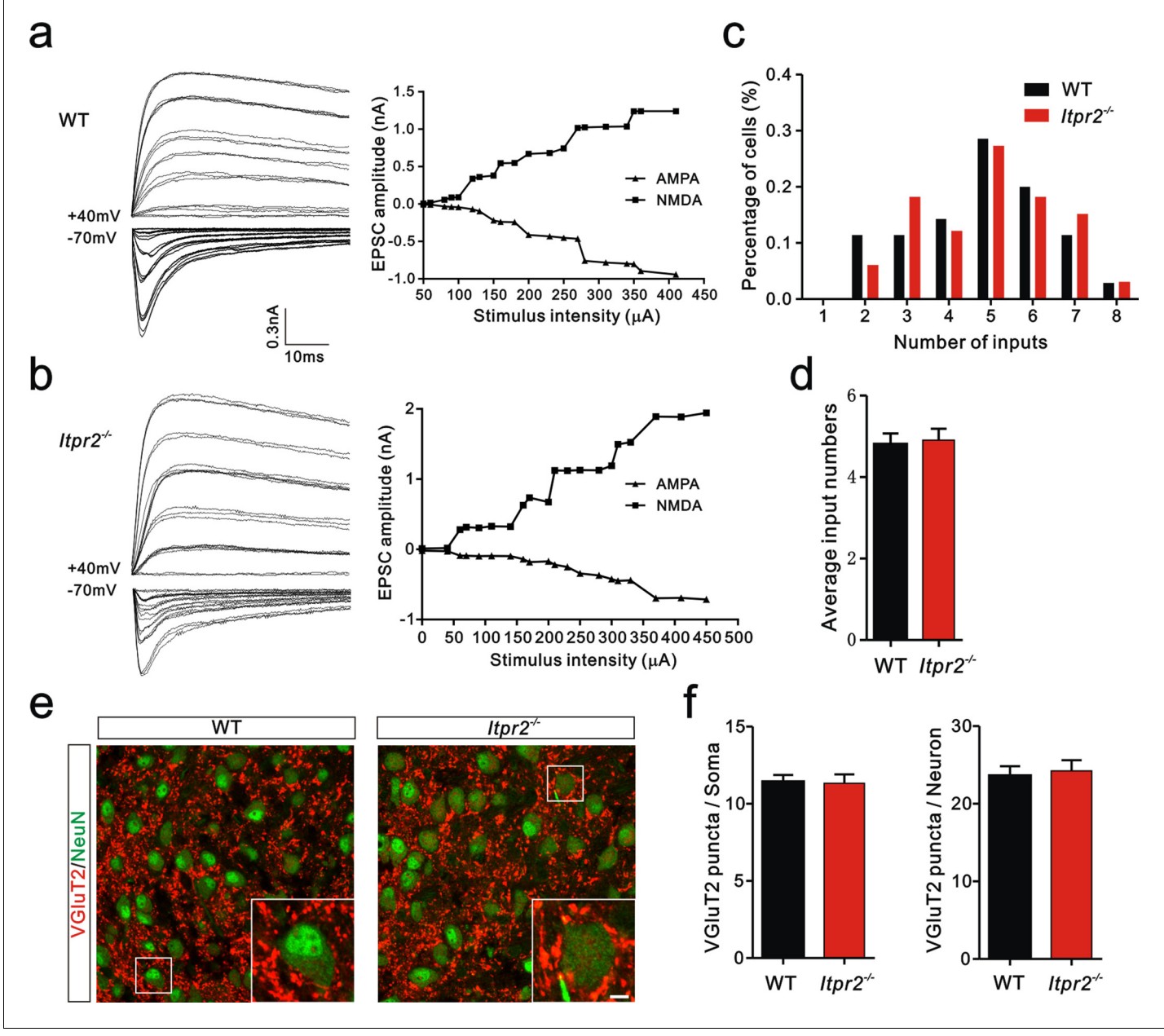

**Figure 2.** Connectivity of Pr5-VPm pathway was comparable in WT and *Itpr2*−/− mice at P7. (a and b) Left panels, sample traces showing membrane current in response to stimuli at a range of intensities in VPm neurons at P7 in WT (a) and *Itpr2*−/− (b) mice. Right panels, peak current *versus* stimulus intensity for WT (a) and *Itpr2*−/− mice (b). (c) Distributions of the number of Pr5 axons innervating each VPm neuron at P7 did not differ between WT (n = 33 cells) and *Itpr2*−/− (n = 35 cells) mice. p=0.78, χ2 test. (d) Histogram of average number of inputs received by each VPm neuron at P7 in WT and *Itpr2*−/− mice (WT = 4.8 ± 0.3, n = 33; *Itpr2*−/− = 4.9 ± 0.3, n = 33). p=0.74, unpaired Student's t test. (e) Sample confocal images of immunostained neurons and Pr5 axonal terminals in the VPm at P7 in WT and *Itpr2*−/− mice. Neurons were visualized with the NeuN antibody (green), and Pr5 axonal terminals were labeled by the VGluT2 antibody (red). Inset is higher-magnification of the boxed area. Scale bar, 5 μm. (f) Quantification of VGluT2 puncta/soma (left, n = 24 cells/group,) and VGluT2 puncta/neuron (right, n = 9 sections from 3 mice/group) for WT and *Itpr2*−/− mice. p>0.05, unpaired Student's t test. Error bars indicate SEM.

The following figure supplement is available for figure 2:

**Figure supplement 1.** Deficient of synapse elimination at P30 in *Itpr2*−/− mice.

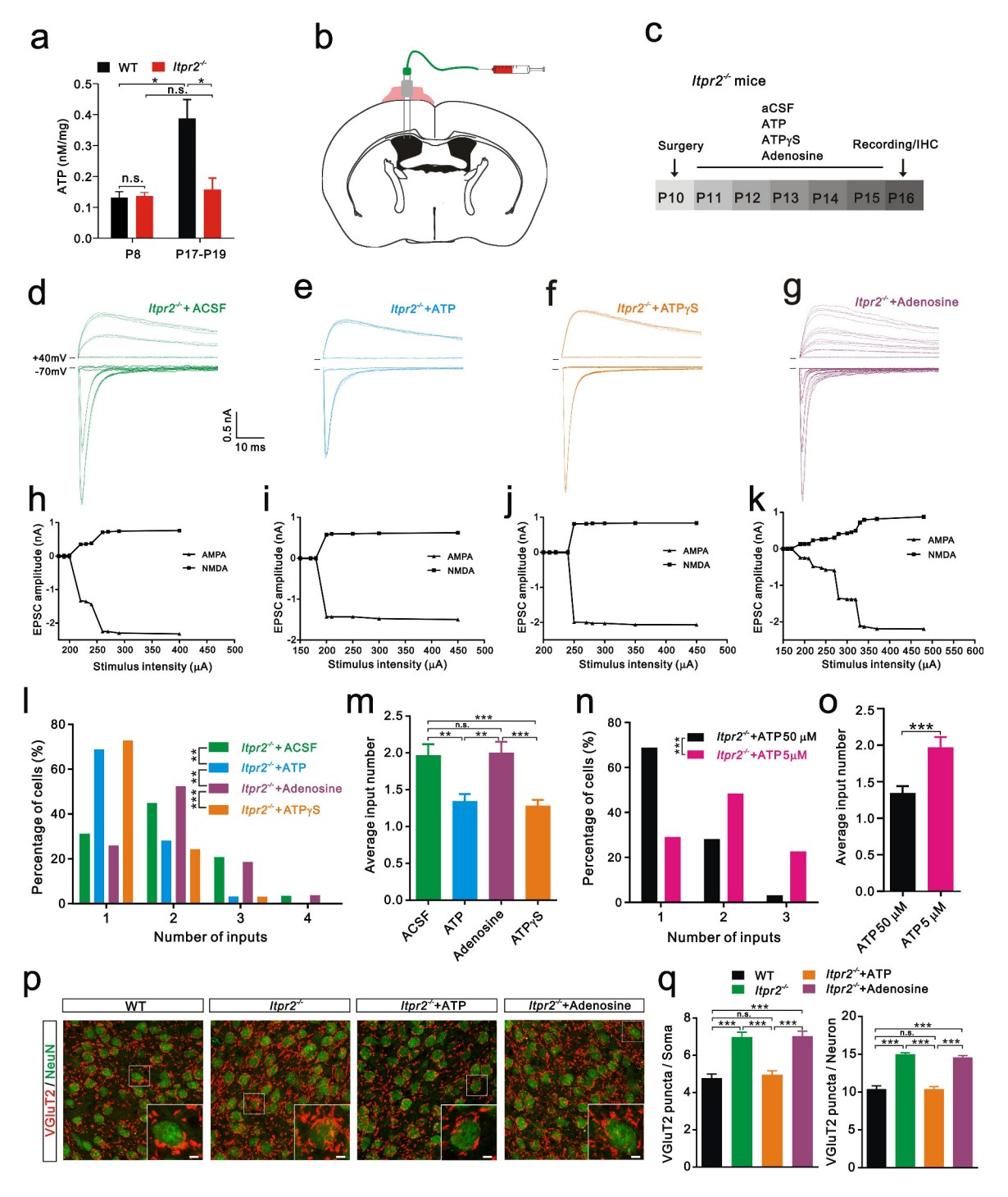

**Figure 3.** Intracerebroventricular injection of ATP from P11 to P15 rescued the synapse elimination deficit in $Itpr2^{-/-}$ mice. (a) Basal ATP levels at P8 and P17-19 in WT and $Itpr2^{-/-}$ mice. *p<0.05, two-way ANOVA followed by Bonferroni *post hoc* test, n = 5 mice per age. n.s, not significant. Error bars indicate SEM. (b) Schematic illustrating the cannula was implanted to the left lateral ventricle. (c) Experimental design of aCSF, ATP (50 μM), ATPγS (50 μM), and adenosine (50 μM) administration and analysis. Injections were given twice per day from P11 to P15. (d-g) Sample traces showing membrane current in response to stimuli with a range of intensities in VPm neurons from $Itpr2^{-/-}$ mice at P16-17 with a short period of aCSF (d), ATP (e), ATPγS (f) and adenosine (g) treatment. (h-k) Peak current *versus* stimulus intensity for aCSF (h), ATP (i), ATPγS (j) and adenosine (k) treatment in $Itpr2^{-/-}$ mice. (l) Distributions of the number of Pr5 axons innervating each VPm neuron at P16-17 with aCSF (n = 29 cells), ATP (n = 32 cells), adenosine (n = 27 cells), and ATPγS (n = 32 cells) treatment in $Itpr2^{-/-}$ mice. **p<0.01, ***p<0.001, $\chi^2$ test. (m) Histogram of average number of inputs received by each VPm neuron at P16-17 in WT and $Itpr2^{-/-}$ mice followed by different injections. ***p<0.001, one-way ANOVA followed by Bonferroni *post hoc*

*Figure 3 continued on next page*

*Figure 3 continued*

test. Error bars indicate SEM. (n) Distributions of the number of Pr5 axons innervating each VPm neuron at P16-17 with different concentrations of ATP treatment in *Itpr2*$^{-/-}$ mice. ***p<0.001, χ$^2$ test. (o) Histogram of average number of inputs received by each VPm neuron at P16-17 in *Itpr2*$^{-/-}$ mice followed by different concentrations of ATP injection. ***p<0.001, unpaired Student's t test. (p) Sample confocal images of immunostained neurons and Pr5 axon terminals in the VPm at P16. Inset is higher-magnification of the white boxed area. Scale bar, 5 μm. (q) Quantification of VGluT2 puncta/soma (left, n = 40 cells/group) and VGluT2 puncta/neuron (right, n = 9 sections from 3 mice/group) for WT, *Itpr2*$^{-/-}$, ATP and adenosine-treated *Itpr2*$^{-/-}$ mice. ***p<0.001, one-way ANOVA. Error bars indicate SEM.

The following figure supplement is available for figure 3:

**Figure supplement 1.** Injury-induced inflammatory responses of astrocytes and microglia are equivalent in aCSF and ATP treated-*Itpr2*$^{-/-}$ mice.

found at P16 was due to a delay in development, we assessed the synapse numbers at P30 by immunostaining for VGluT2. We found that even at P30, there were more Pr5-VPm synapses in *Itpr2*$^{-/-}$ mice than in WT mice (*Figure 2—figure supplement 1a,b*). Therefore, knockout of IP3R2 disrupted developmental synapse elimination at the VPm relay synapses.

## Intracerebroventricular injection of ATP from P11 to P15 rescued the deficit in synapse elimination in *Itpr2*$^{-/-}$ mice

Increasing evidence suggests that astrocytes release a number of gliotransmitters, such as glutamate, ATP, and D-serine to regulate synaptic transmission and synaptic plasticity (*Chen et al., 2013*; *Henneberger et al., 2010*; *Jourdain et al., 2007*). Among these gliotransmitters, we were particularly interested in ATP and assumed that it might play a role in synapse elimination for the following reasons: 1) the basal ATP levels are reduced in *Itpr2*$^{-/-}$ mice (*Cao et al., 2013*); 2) astrocytes release ATP in a Ca$^{2+}$-dependent manner (*Lalo et al., 2014*; *Zhang et al., 2007*); and 3) ATP regulates synaptic plasticity (*Chen et al., 2013*). We then tested whether the basal ATP level in the VPm changes during development at the time when exploratory activity increases in WT mice. A significant increase in the basal ATP level occurred at P18 compared with that at P7 in WT mice (*Figure 3a*). This developmental up-regulation of the basal ATP level was absent in *Itpr2*$^{-/-}$ mice (*Figure 3a*). In addition, we found that the basal ATP level was comparable between WT and *Itpr2*$^{-/-}$ mice at P7 but was significantly reduced in KO mice at P18 (*Figure 3a*). These results suggested that, during development, the basal ATP level increases in WT but not in *Itpr2*$^{-/-}$ mice. Next, we tested whether the deficit of synapse elimination in *Itpr2*$^{-/-}$ mice could be rescued by compensatory ATP. To achieve this goal, we implanted a cannula in the left lateral ventricular of the brain (*Figure 3b*) and found that intracerebroventricular injection of ATP from P11 to P15 rescued the synapse elimination deficit (*Figure 3e,i,l,m*). At P16-17, unlike aCSF-treated mice, the majority of VPm neurons in *Itpr2*$^{-/-}$ mice that had received ATP (50 μM) treatment was innervated by a single Pr5 input (*Figure 3l*). Immunostaining for VGluT2 also revealed that the number of synapses significantly decreased in *Itpr2*$^{-/-}$ mice with ATP treatment (*Figure 3p,q*). These results indicate that ATP treatment is sufficient to rescue synapse elimination deficit in *Itpr2*$^{-/-}$ mice and the removal of redundant synapses is independent of IP3R2-dependent Ca$^{2+}$ signaling. Cannula implantation caused injury may induce inflammatory responses of astrocytes and microglia differentially between aCSF and ATP treated-*Itpr2*$^{-/-}$ mice, thereby affected the rescue effects. We found that reactive astrocytes and activated microglia around the site of cannula placement in aCSF and ATP treated-*Itpr2*$^{-/-}$ mice were identical (*Figure 3—figure supplement 1*), and ruled out this possibility. We next applied a low dose ATP (5 μM) and found that the impaired synapse elimination cannot be rescued in *Itpr2*$^{-/-}$ mice, suggesting the rescue effect of ATP was in a dose-dependent manner (*Figure 3n,o*).

Given that ATP is readily hydrolyzed, and to exclude the possibility that the degradation products of ATP play a role in synapse elimination, we injected the non-hydrolyzable ATP analog ATPγS and found that, ATPγS (50 μM) also rescued the synapse elimination impairment in *Itpr2*$^{-/-}$ mice (*Figure 3f,j,l,m*). These results suggested that astrocytes mediate synapse elimination *via* the Ca$^{2+}$-dependent release of ATP. Since adenosine is a degradation product of ATP and could modulate synaptic plasticity by activating adenosine receptors (*Pascual et al., 2005*), we set out to exclude the possibility that adenosine rescued the synapse elimination deficit in *Itpr2*$^{-/-}$ mice. We found intracerebroventricular injection of adenosine (50 μM) throughout P11-P15 had no effect on Pr5-

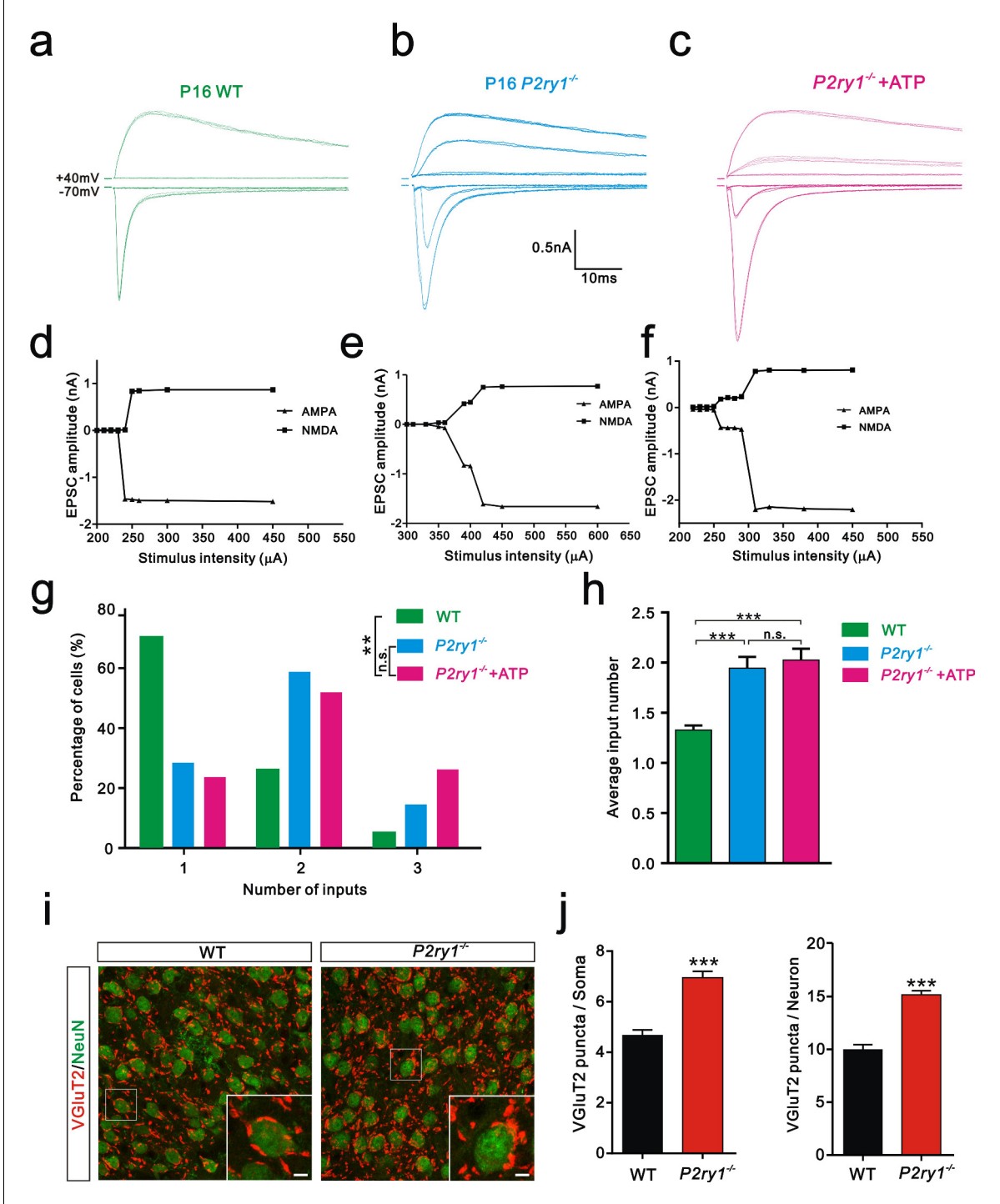

**Figure 4.** Synapse elimination was also impaired in *P2ry1*−/− mice at P16-17 and cannot be rescued by ATP. (a-c) Sample traces of membrane current in response to stimuli at a range of intensities in VPm neurons at P16-17 in WT (a), *P2ry1*−/− (b), and *P2ry1*−/− with ATP injection (c) mice. (d-f) Peak current *versus* stimulus intensity for WT (d), *P2ry1*−/− (e), and *P2ry1*−/− with ATP injection (f) mice. (g) Distributions of the number of Pr5 axons innervating each VPm neuron at P16-17 in WT (n = 33 cells), *P2ry1*−/− (n = 47 cells) and *P2ry1*−/− with ATP injection (n = 39 cells) mice. **p<0.01, χ2 test. (h) Average number of Pr5 axonal inputs received by individual VPm relay neurons significantly increased in *P2ry1*−/− mice compared to WT controls. ***p<0.001 by one-way ANOVA. (i) Sample confocal images of immunostained neurons and Pr5 axon terminals in the VPm at P16. Neurons were visualized with the NeuN antibody (green), and Pr5 axon terminals were labeled by the VGluT2 antibody (red). Inset is higher-magnification of the boxed area. Scale bar, 5 μm. (j) Quantification of VGluT2 puncta/soma (left, n = 42 cells/group) and VGluT2 puncta/neuron (right, n = 9 sections from 3 mice/group) in WT and *P2ry1*−/− mice. ***p<0.001, unpaired Student's t test. Error bars indicate SEM.
*Figure 4 continued on next page*

*Figure 4 continued*

The following figure supplements are available for figure 4:

**Figure supplement 1.** Expression pattern of P2Y1 receptors in the VPm.

**Figure supplement 2.** Calcium imaging in hippocampus and the VPm of WT and *P2ry1*$^{-/-}$ mice.

**Figure supplement 3.** The number of inputs received by each VPm neuron at P7 is equivalent between WT and *P2ry1*$^{-/-}$ mice.

VPm connectivity at P16 in these mice (*Figure 3g, k, l, m*). Thus, the synapse elimination deficit in *Itpr2*$^{-/-}$ mice was rescued by ATP, but not adenosine.

## ATP modulates synapse elimination through activation of P2Y1 receptors

Having shown that ATP rescued the synapse elimination deficit at the VPm relay synapses in *Itpr2*$^{-/-}$ mice, the next question is how this process occurs. In the CNS, ATP could activate purinegic signaling through P2X or P2Y receptors to modulate synapse function. Particularly, P2Y1 receptors have been shown participated in ATP-induced heterosynaptic long-term depression (LTD) in hippocampus (*Chen et al., 2013*), and LTD has been suggested correlated to synapse elimination (*Bastrikova et al., 2008*; *Wiegert and Oertner, 2013*). To test the possibility of involvement of P2Y1 receptors in synapse elimination at the VPm, we first examined the expression pattern of P2Y1 receptors and found that unlike in the hippocampus (*Zhu and Kimelberg, 2004*), they were only expressed in neurons in the VPm (*Figure 4—figure supplement 1*). We next found that ATP-induced

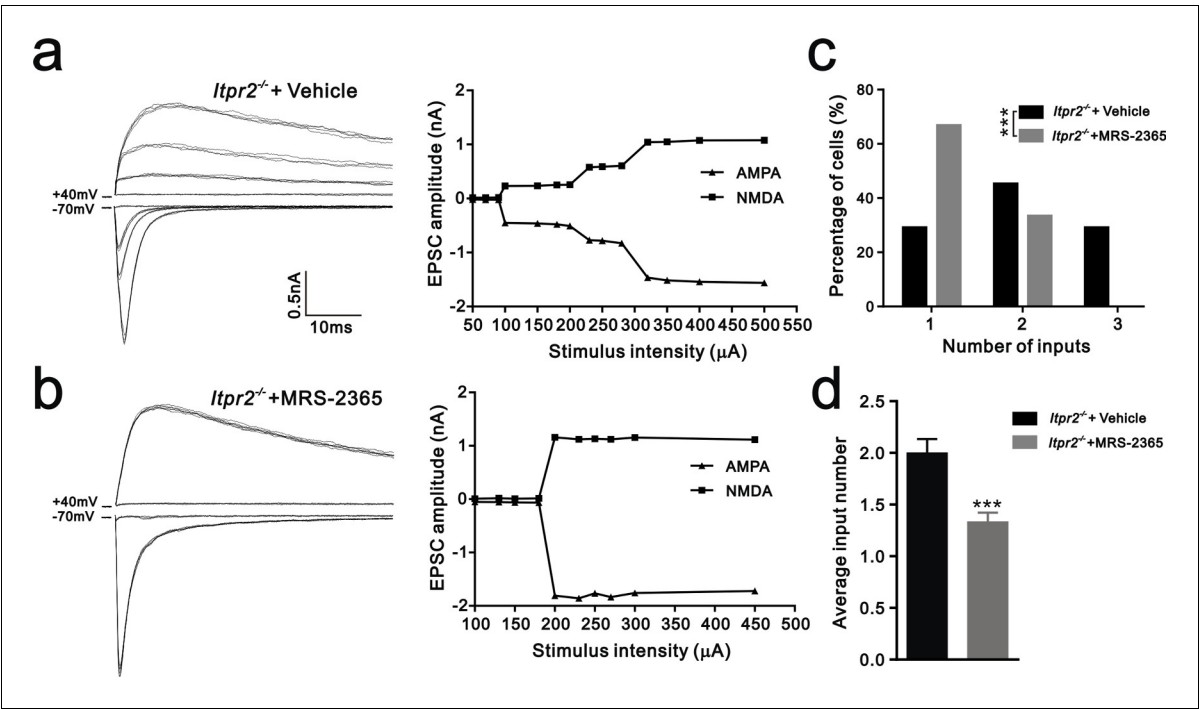

**Figure 5.** Synapse elimination was rescued by P2Y1 receptor agonist MRS-2365 in *Itpr2*$^{-/-}$ mice. (a and b) Left panels, sample traces of membrane current in response to stimuli over a range of intensities in VPm neurons at P16-17 in aCSF (a) and MRS-2365 (b) treated-*Itpr2*$^{-/-}$ mice. Right panels, peak current *versus* stimulus intensity for aCSF (a) and MRS-2365 (b) treated-*Itpr2*$^{-/-}$ mice. (c) Distributions of the number of Pr5 axons innervating each VPm neuron during P16-17 in aCSF (n = 32 cells) and MRS-2365 treated-*Itpr2*$^{-/-}$ mice(n = 30 cells). ***p<0.001, $\chi^2$ test. (d) Histogram of average number of inputs received by each VPm neuron at P16-17 in aCSF (n = 32 cells) and MRS-2365 (n = 30 cells) treated-*Itpr2*$^{-/-}$ mice. ***p<0.001, unpaired Student's t test. Error bars indicate SEM.

somatic $Ca^{2+}$ elevation in astrocytes was impaired in hippocampus, but not in the VPm in $P2ry1^{-/-}$ mice (*Figure 4—figure supplement 2*), further indicated that the expression pattern of P2Y1 receptors might be different in hippocampus when compared with that in the VPm. In line with immunostaining results, we also found that ATP-induced $Ca^{2+}$ elevation in VPm neurons of $P2ry1^{-/-}$ mice was impaired (*Figure 4—figure supplement 2*). These results suggested that even in the VPm, the expression pattern of P2Y1 may have specificity among different cell types. Next, we determined whether synapse elimination was disrupted in $P2ry1^{-/-}$ mice. Compared to WT mice, the majority of VPm neurons in $P2ry1^{-/-}$ mice received multiple Pr5 inputs at P16 (WT = 30%, 13 of 33 cells; $P2ry1^{-/-}$ = 72%, 34 of 47 cells, *Figure 4g*). Consistent with the electrophysiological results, we confirmed the synapse elimination deficit in $P2ry1^{-/-}$ mice with immunostaining for VGluT2 (*Figure 4i*). There was more VGluT2 staining around the somata of VPm neurons in $P2ry1^{-/-}$ mice than in WT mice (*Figures 4i,j*). Therefore, synapse elimination was also impaired in $P2ry1^{-/-}$ mice. To address the abnormal of synaptic connectivity in $P2ry1^{-/-}$ mice at P16 was actually due to developmental failure, we tested Pr5-VPm synapses at P7. We found that the connectivity of Pr5-VPm pathway was comparable between WT and $P2ry1^{-/-}$ mice at P7 (*Figure 4—figure supplement 3*). In addition, the synapse elimination deficit in these mice was not rescued by intracerebroventricular injection of ATP from P11 to P15 (*Figure 4c, f, g, h*). These results suggested that synapse elimination promoted by ATP may depend on P2Y1 receptors. If this is the case, one direct evidence would be the rescue of the synapse elimination defect with application of the P2Y1 agonist in $Itpr2^{-/-}$ mice. We next found that the deficit of synapse elimination was rescued by intracerebroventricular injection of a selective P2Y1 receptors agonist, MRS-2365 (20 µM) from P11 to P15 (*Figure 5*). Unlike aCSF treatment, the majority of VPm neurons in $Itpr2^{-/-}$ mice with treatment of MRS-2365 innervated by a single Pr5 input (*Figure 5c*). The mean number of inputs received by each VPm neuron was significantly decreased in $Itpr2^{-/-}$ mice with intracerebroventricular injection of MRS-2365 (2.0 ± 0.76 MRS-2365 versus 1.33 ± 0.48 aCSF, *Figure 5d*). Taken together, these data confirmed that ATP promotes synapse elimination *via* P2Y1 receptors.

## Discussion

Here, by using electrophysiological recording in transgenic mice, we found that astrocytes contribute to synapse elimination through $Ca^{2+}$-dependent release of ATP and activation of P2Y1 receptors. Specifically, we demonstrated that (1) developmental synapse elimination in the VPm was impaired in the $Itpr2^{-/-}$ mice; (2) intracerebroventricular injection of ATP and ATPγS, but not adenosine, from P11 to P15, rescued the synapse elimination deficit in $Itpr2^{-/-}$ mice; (3) $P2ry1^{-/-}$ mice also showed impaired synapse elimination and this could not be rescued by ATP; and (4) deficient of synapse elimination in $Itpr2^{-/-}$ mice was rescued by intracerebroventricular injection of MRS-2365, a selective P2Y1 receptors agonist, from P11 to P15.

In the CNS, neurons make redundant synaptic connections early in development, and most of inappropriate synapses are eliminated later to form precise neural circuitry (*Lichtman and Colman, 2000*; *Luo and O'Leary, 2005*). This process is believed to largely rely on neuronal activity, while the underlying mechanisms are still unclear. Recent studies have shown that astrocytes show a transient $[Ca^{2+}]_i$ elevation, which depends on the GPCR/IP3R pathway, in response to neuronal activity and subsequently release gliotransmitters to modulate synaptic transmission and plasticity (*Araque et al., 2014*). Therefore, astrocytic $[Ca^{2+}]_i$ elevation is a consequence of neuronal activity. Based on these reasons, we proposed that activity-dependent competition for synapse elimination involves both postsynaptic neuronal activation and astrocytic $[Ca^{2+}]_i$ elevation. A large literature has shown that GPCR/IP3R-dependent $Ca^{2+}$ signaling is critical for astrocytes to release gliotransmitters and modulate synaptic transmission. Conversely, recent studies from McCarthy's laboratory have suggested that IP3R2-mediated $Ca^{2+}$ signaling in astrocytes may not be involved in the acute modulation of neuronal activity and synaptic transmission (*Fiacco et al., 2007*; *Petravicz et al., 2008*). Two studies have shown that there were still some $Ca^{2+}$ fluctuations occurring in astrocytes process, although the somatic $Ca^{2+}$ signaling was eliminated in $Itpr2^{-/-}$ mice (*Kanemaru et al., 2014*; *Srinivasan et al., 2015*). These results suggested that $Ca^{2+}$ signaling was not totally removed in these mutants. In line with previous studies, we found that the lack of ATP-induced $Ca^{2+}$ increase in astrocytes soma in these mutants. Our studies strongly suggested that astrocytic IP3R2-dependent somatic $Ca^{2+}$ signaling is required for developmental synapse elimination. Considering the dynamic

and complexity, the astrocytic Ca$^{2+}$ signaling in the process and the soma may play different roles in gliotransmitter release and regulating neuronal functions (*Volterra et al., 2014*).

It is important to note that synapse elimination was only partly impaired in the *Itpr2*$^{-/-}$ and *P2ry1*$^{-/-}$ mice. Although we found both anatomical and electrophysiological evidence that these mice had significant deficits in developmental synapse elimination, some degree of synaptic pruning still occurred during P7 to P16, suggesting that there are multiple mechanisms for controlling synapse elimination. Indeed, synaptic pruning in the VPm relay synapses occurs at several developmental stages (*Wang and Zhang, 2008*). Before P10, pruning is insensitive to whisker deprivation, but then later, the elimination of redundant inputs is largely dependent on somatosensory experience. In the mutants, such as *Itpr2*$^{-/-}$ mice, the nature of sensory information transmitted from whisker to the VPm might be altered, thus causes the defects of synapse elimination. We found surprisingly, the connectivity between Pr5 and VPm in *Itpr2*$^{-/-}$ mice undergone ATP and ATPγS treatments restored to WT mice level. These results suggested that no matter sensory information had been changed or not, ATP was sufficient to rescue the impairment of synapse elimination in *Itpr2*$^{-/-}$ mice. However, the rescue effect of ATP may occur at a site between the whisker and VPm, and this possibility could not be ruled out by our study.

It is about P13 that mice open their eyes and start to use their vibrissae to explore the environment. Accompanying this behavioral change, we found a significant enhancement of the basal ATP level in the VPm of WT mice between P7 and P18 (*Figure 3a*). However, the basal ATP level failed to be up-regulated in *Itpr2*$^{-/-}$ mice during development. In line with previous studies (*Cao et al., 2013*), we also found a decreased basal ATP level in the brain of *Itpr2*$^{-/-}$ mice. ATP can be released by both neurons and glia. The decreased ATP level in *Itpr2*$^{-/-}$ mice was most likely due to astrocytes since IP3R2 was not expressed in neurons and [Ca$^{2+}$]$_i$ elevation did not change in neurons. Although ATP itself is a signaling molecule by activating P2X or P2Y receptors, the degraded form adenosine is also a ligand for adenosine receptors and causes synaptic depression (*Wu and Saggau, 1994*). The fact that adenosine injection through P11-15 did not rescue the synapse elimination deficit in *Itpr2*$^{-/-}$ mice excludes the possibility of a role for adenosine in synapse elimination. In addition to being a signal molecule, ATP is also energy currency in the brain and the phenotype of *Itpr2*$^{-/-}$ mice could be a result of global metabolic stress. This possibility is very unlikely because either ATPγS, a non-hydrolyzable ATP analog or one selective P2Y1 receptors agonist MRS-2365 could rescue the deficit of synapse elimination in *Itpr2*$^{-/-}$ mice. In our study, we concluded that ATP was not acting as an energy supply to play a role in synapse elimination.

Although both of *Itpr2*$^{-/-}$ and *P2ry1*$^{-/-}$ mice showed deficits of developmental elimination of relay synapses at the VPm, it is not necessary that they are mechanistically related. That is, the phenotypes in these two mutants are two independent events. The results of P2Y1 receptors agonist injection rescued the deficient of synapse elimination in *Itpr2*$^{-/-}$ mice provided a direct linkage between IP3R2-dependent releasing of ATP and its downstream purinergic pathway. Unlike IP3R2, studies have shown that P2Y1 receptors are widely expressed in astrocytes, microglia and neurons. Thus, loss function of P2Y1 receptors in all three types of cell may contribute to the phenomenon of impaired synapse elimination in *P2ry1*$^{-/-}$ mice. First, P2Y1 receptors have been shown to be expressed by astrocytes (*Pascual et al., 2012*), therefore the finding that *P2ry1*$^{-/-}$ mice has impaired synapse elimination could be due to a failure to increase calcium levels in astrocytes in response to ATP. This possibility could be excluded by the results of astrocytes in the VPm of *P2ry1*$^{-/-}$ mice responded to ATP with a similar increase in intracellular Ca$^{2+}$ when compared to cells in the WT mice (*Figure 4—figure supplement 2*). Additionally, we found that P2Y1 receptors were not expressed by astrocytes in the VPm (*Figure 4—figure supplement 1a*), but abundantly expressed in the hippocampus astrocytes (data not shown). Second, P2Y1 receptors have been reported to be expressed in microglia and ATP activates microglia. The phenotype of synapse elimination defects in *P2ry1*$^{-/-}$ mice could be explained by injection of ATP activates microglia-mediated synapse elimination. At present, there were some mRNA and functional clues showing that microglia expressed P2Y1 receptors, whereas the protein evidence was very few (*Fields and Burnstock, 2006*). In our study, we did not find any P2Y1 protein expression on microglia in both of hippocampus (data not shown) and the VPm (*Figure 4—figure supplement 1b*). Therefore, the protein level of P2Y1 receptors expressed on microglia in the VPm, if any, was very low. However, we still could not totally exclude the possibility that microglia may contribute to synapse elimination through P2Y1 receptors. Further studies by using cell-type specific knockout mice are needed to answer this

question. Last, ATP-mediates synapse elimination may occur through activation of neuronal P2Y1 receptors. This speculation is most likely due to P2Y1 receptors are abundantly expressed in VPm neurons (*Figure 4—figure supplement 1c*). In addition, our $Ca^{2+}$ imaging data showing that ATP-evoked $Ca^{2+}$ response in VPm neurons was significantly decreased in *P2ry1$^{-/-}$* mice. Our findings strongly suggest that ATP may activate neuronal P2Y1 receptors to promote synapse elimination. This hypothesis is supported by a previous study showing that activation of neuronal P2Y1 receptors is required for astrocytes derived ATP mediated LTD in hippocampus (*Chen et al., 2013*). Due to a close linkage proposed between LTD and synapse elimination (*Wiegert and Oertner, 2013*), one possible model is that ATP and downstream purinergic signaling could recognize unwanted synapses. Certainly, further studies are needed to elucidate the whole picture of pathways that how activation of purinergic signal mediating synapse elimination.

## Materials and methods

### Animals
IP3R2-knockout (*Itpr2$^{-/-}$*) mice were a kind gift from Dr. Ju Chen (University of California, San Diego) and maintained as a heterozygous line. Heterozygous (*Itpr2$^{+/-}$*) mice were interbred to generate homozygous full-mutant mice (*Itpr2$^{-/-}$*) and WT littermate controls (*Itpr2$^{+/+}$*), which were used in experiments. *Cx3cr1$^{GFP/+}$* mice,hGFAP-GFP mice, and heterozygous *P2ry1$^{-/-}$* mice were obtained from Jackson Laboratories.

### Slice preparation
Sagittal slices were obtained as previously described (*Wang and Zhang, 2008*). Mice and littermate controls were anesthetized with sodium pentobarbital, and then perfused with ice-cold oxygenated slicing solution. The brain was removed rapidly and immersed in ice-cold slicing solution containing (in mM): 110 choline chloride, 7 $MgCl_2 \cdot 6H_2O$, 2.5 KCl, 0.5 $CaCl_2 \cdot H_2O$, 1.3 $NaH_2PO_4$, 25 $NaHCO_3$, 20 glucose, saturated with 95% $O_2$ and 5% $CO_2$. The brain was cut into 300 µm slices on a vibratome (MicromHM650V). Slices were allowed to recover for 40 min at 32°C and then at room temperature for recording in artificial cerebrospinal fluid (aCSF) containing (in mM): 125 NaCl, 2.5 KCl, 2 $CaCl_2 \cdot H_2O$, 1.3 $MgCl_2 \cdot 6H_2O$, 1.3 $NaH_2PO_4$, 25 $NaHCO_3$, 10 glucose. Oxygen was continuously supplied during recovery and recording. To block inhibitory synaptic transmission, picrotoxin (100 µM) was added to the bath.

### Electrophysiology
All recordings were made at room temperature in a submerged recording chamber with constant aCSF perfusion. Patch electrodes had a resistance of 2–4 MΩ when filled with an internal solution containing (in mM): 110 Cs methylsulfate, 20 TEA-Cl, 15 CsCl, 4 ATP-Mg, 0.3 GTP, 0.5 EGTA, 10 HEPES, 4.0 QX-314, and 1.0 spermine (pH adjusted to 7.2 with CsOH, 290–300 mOsm with sucrose). Whole-cell voltage-clamp recordings were made from the soma of VPm neurons with an Axopatch 200B amplifier and Digidata 1322A with pCLAMP 8.1 software (Molecular Devices). Cells were visualized under an upright microscope (BX51WI, Olympus) with infrared optics. Signals were filtered at 2 kHz and digitized at 10 kHz. The series resistance (Rs) was <20 MΩ with no compensation and data were discarded when this varied by $\geq$ 20%.

To determine the numbers of inputs to a VPm neuron, we recorded evoked EPSCs from the same cells at holding potentials of -70 and +40 mV cell over a wide range of stimulus intensity. A concentric electrode (World Precision Instruments) was placed on the medial lemniscal fiber bundle and stimuli (typically from 20 to 600 µA, 100 µs) were delivered at 0.1 Hz *via* a Grass S88 stimulator. To distinguish lemniscal synaptic responses from corticothalamic responses, paired-pulse stimulation with an interval of 100 ms was used. First, we used increments of 50–100 µA to search for step numbers. Then we used small increments of 1–10 µA near each transition point to ensure that it was actually a single step. After that, we applied a stronger stimulus, the intensity of which was at least twice than the last step.

## Extracellular ATP measurement

The concentration of extracellular ATP was determined with a bioluminescent ATP assay kit (Sigma), as previously described (*Zhang et al., 2007*). The ectonucleotidase inhibitor dipyridamole (10 μM) was added to the extracellular solution throughout the experiment to decrease ATP hydrolysis. Luminescence was measured with a luminometer (Varioskan Flash, Thermo) according to the manufacturer's instructions. A calibration curve was constructed from standard ATP samples and the luminescence of the incubated medium was measured as the background ATP level. To measure the ATP content in acute VPm slices, they were incubated in oxygenated ACSF for 10 min at room temperature. The ACSF was then collected for ATP assay. For quantification, the amount of protein was used for normalization and was determined using the Enhanced BCA protein assay kit (Beyotime, China).

## Immunostaining

Animals were anesthetized with sodium pentobarbital, and perfused with 0.9% saline followed by ice-cold 4% paraformaldehyde. Brains were removed and post-fixed overnight in 4% paraformaldehyde at 4°C and then transferred to 30% sucrose in PBS at 4°C for 2 days. Sagittal sections 30 or 40 μm thick were cut on a microtome (Leica CM 1850). Sections were dried, washed three times in 0.01M PBS and with 0.3% Triton-100X in 0.1M PBS (40 min) or frozen methanol (10 min at -20°C), then blocked with 10% BSA for 1 hr at room temperature. Sections were incubated with primary antibodies as follows: VGluT2 (guinea-pig polyclonal, 1:2000, Millipore), NeuN (monoclonal mouse, 1:500; Millipore), GFAP (rabbit, 1:500, Chemicon), GFAP (monoclonal mouse, 1:500; Synaptic System), Iba1 (polyclonal rabbit, 1:200; Chemicon), IP3R2 (polyclonal rabbit, 1:200; Santa Cruz), P2Y1 (polyclonal rabbit, 1:200; Abcom) at 4°C for 12–24 hr. Secondary Alexa-conjugated antibodies were added at 1:1000 in 0.1 M PBS for 2 hr at room temperature. Images were captured using an Olympus FV-1200 inverted confocal microscope.

## Brain slice Ca$^{2+}$ imaging

In brief, for Ca$^{2+}$ imaging in hippocampus, WT, $Itpr2^{-/-}$ mice, and $P2ry1^{-/-}$ (P13-P16) were sacrificed and coronal slices of hippocampus were cut and incubated in ACSF at room temperature, saturated with 95% O$_2$ and 5% CO$_2$. For Ca$^{2+}$ imaging in VPm, sagittal slices containing VPm were cut. Slices were incubated with 10 μM Fluo-4/AM (Invitrogen) or Cal-250 (AATBioquest) for 45 min at room temperature in oxygenated ACSF. In these conditions, most of the loaded cells in the stratum radiatum and VPm were astrocytes as confirmed by SR101 staining. CA1, dentate gyrus, and VPm neurons were also loaded in these conditions. Before imaging, slices were transferred to stain-free ACSF for at least 10 min to wash out the stain. Images were acquired every 3 s using an Olympus FV-1200 confocal microscope. Average fluorescence intensity was measured from analysis boxes over the cell-bodies of astrocytes and neurons. When the baseline was stable, ATP (100 μM) was added to evoke intracellular Ca$^{2+}$ release from astrocytes and neurons. Increases in the fluorescence intensity over baseline were calculated for each trace and are reported as ΔF/F$_0$.

## Quantification of VGluT2

Confocal images were collected with a 60× objective (NA 1.3, water) on an Olympus FV-1200 inverted confocal microscope. Z-stacks were obtained by 10 consecutive steps with 0.5 μm interval thickness. To determine the VGluT2 puncta on the soma of a thalamic VPm neuron, data were opened with Fluoview software (Olympus) and then exported as 8-bit tiff files. Neurons were randomly selected and VGluT2 puncta were counted manually using the Cell Counter plugin of ImageJ. In most cases, the VGlut2 puncta are separated from each other. Sometimes there are multiple puncta that stay very close, but we can still discriminate them by the little gaps among them. Occasionally, there is a big puncta that has no gap, which is then referred as one puncta. We count the puncta that contact the NeuN as puncta/soma. To quantify the average VGluT2 puncta per neuron, the density of VGluT2 puncta was analyzed using the Analyze Particles command in ImageJ (version 1.43). Thalamic VPm neuron number was counted manually using the Cell Counter plugin of ImageJ. The puncta/neuron is then determined by all the puncta number divided by all neuron number in a given image.

## Neuron density quantification in the VPm

For quantification of neuron density, 9 sections from 3 different animals (3 sections per animal) were used. The VPm was acquired by using a 10×objective (NA 0.4). Thalamic VPm neuron numbers were counted manually by using the Cell Counter plugin of Image J. To calculate the density of neurons, only the area of the VPm was measured.

## Cannula implantation and pharmacological experiments

Mice at P10 were deeply anaesthetized with isoflurane and placed into a stereotactic apparatus. A 30-gauge stainless steel infusion cannula (RWD life science) was implanted unilaterally in the left lateral ventricle (-0.4 mm from bregma, 0.9 mm lateral from midline, and 2.5 mm vertical from the cortical surface) and was fixed to the skull with dental cement. After one day recovery, drugs including aCSF, ATP (5 or 50 µM, 2 µl), adenosine (50 µM, 2 µl), ATPγS (50 µM, 2 µl), MRS-2365 (20 µM, 2 µl) were delivered to lateral ventricle slowly by using a microsyringe pump which was connected to the infusion cannula by a PE20 tube. Injections were given twice per day at the interval of 12 hr from P11 to P15. After experiments, the location of the cannula was verified histologically and discarded the animal which did not target to lateral ventricle.

## Inflammatory response quantification

To analysis the inflammatory responses of astrocytes and microglia, fixed sections were immunolabeled for GFAP and Iba1. For quantification the maximum distance of reactive cells to the injury site, three images were collected from each animal on a confocal microscopy by using 1 µm z-step. Astrocytes in the cortex express a low level of GFAP in normal situation. They were very sensitive to injury as indicated by an increased GFAP expression. Therefore, the distance of GFAP positive cells to injury site was measured as the distance of inflammatory responses. For quantification of reactive astrocyte, the number of GFAP positive cells within 150 µm from the injury site was counted. For quantification of microglia activation, the number of Iba1[+] microglia within 150 µm from the injury site was counted. Then the percentage of amedoid-like activated microglia was calculated.

## Statistical analysis

All results are presented as mean ± SEM. GraphPad Prism 5 (La Jolla, CA) software was used for all statistical analyses. Differences between two groups were evaluated by un-paired or paired Student's t-test. For multi-group comparisons, one-way ANOVA with Tukey's multiple-comparisons test was used. Two-way ANOVA followed by the Bonferroni *post hoc* test was used for *Figure 3A*. Frequency distributions were analyzed with $\chi^2$ test. The significance level was set at $p < 0.05$.

## Acknowledgement

We thank Dr. Zhong-wei Zhang, Dr. Da-ting Lin, Dr. Xiang-yao Li, Dr. Jia-dong Chen, and Dr. IC Bruce for critical comments on this manuscript. This work was supported by grants from the Major State Basic Research Program of China (2011CB504400, 2013CB945600, 2015CB755600, 2015AA020515), the National Natural Science Foundation of China (31471022, 81221003, 91232000, 31490590), the Program for Introducing Talents in Discipline to Universities, the Zhejiang Provincial Natural Science Foundation of China (Y2110057), and the Fundamental Research Funds for the Central Universities (2014FZA7007).

## Additional information

### Funding

| Funder | Grant reference number | Author |
| --- | --- | --- |
| National Natural Science Foundation of China | 31471022 | Shumin Duan Hao Wang |
| Major State Basic Research Program of China | 2011CB504400 | Shumin Duan Hao Wang |

| Major State Basic Research Program of China | 2013CB945600 | Shumin Duan Hao Wang |
| Major State Basic Research Program of China | 2015CB755600 | Shumin Duan Hao Wang |
| Major State Basic Research Program of China | 2015AA020515 | Shumin Duan Hao Wang |
| The Program for Introducing Talents in Discipline to Universities | | Shumin Duan |
| The Fundamental Research Funds for the Central Universities | 2014FZA7007 | Shumin Duan |
| Zhejiang Provincial Natural Science Foundation of China | Y2110057 | Shumin Duan Hao Wang |
| National Natural Science Foundation of China | 81221003 | Shumin Duan Hao Wang |
| National Natural Science Foundation of China | 91232000 | Shumin Duan Hao Wang |
| National Natural Science Foundation of China | 31490590 | Shumin Duan Hao Wang |

The funders had no role in study design, data collection and interpretation, or the decision to submit the work for publication.

## Author contributions

JY, HY, Drafting the article, Final approval of the version to be published, Conception and design, Acquisition of data, Analysis and interpretation of data; YL, Discussing the revision of the article, Final approval of the version to be published, Acquisition of data, Analysis and interpretation of data; XL, LQ, HL, Discussing the revision of the article, Final approval of the version to be published, Contributed unpublished essential data or reagents; SD, Discussing the revision of the article, Final approval of the version to be published, Conception and design; HW, Conception and design, Analysis and interpretation of data, Drafting or revising the article

## Author ORCIDs

Hao Wang, http://orcid.org/0000-0003-1862-8229

## Ethics

Animal experimentation: Animal experiments were conducted in accordance with the Guidelines for the Care and Use of Laboratory Animals of Zhejiang University, and were approved by Committee of Laboratory Animal Center of Zhejiang University (ZJU201501402). All surgery was performed under sodium pentobarbital anesthesia, and every effort was made to minimize suffering.

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
