## [Decision Letter]

Thank you for resubmitting your work entitled "Astrocytes contribute to synapse elimination via type 2 inositol 1,4,5-trisphosphate receptor-dependent release of ATP" for further consideration at *eLife*. Your article has been favorably evaluated by Gary Westbrook (Senior editor) and two reviewers. This study is interesting and informative as it identifies molecular mechanisms important to synapse refinement. In the revision, the authors have removed the problematic ATP induced LTD model, improved the quality of their images and rewritten their Discussion section. The manuscript is much improved with these changes. There are just a few remaining suggestions to be addressed:

1) It should be made very clear in the figure legend of Figure 1—figure supplement 1 that the co-expression data of IP3R2 was done in the hippocampus, and that the authors could not obtain similar data in VPm because the same antibodies that worked for hippocampus did not work for thalamus.

2) In the third paragraph of the Discussion: The authors argue against the possibility that their synapse elimination findings may arise from altered presynaptic activity because ATP and ATPγS rescue the phenotype. However, they cannot rule out that this rescue occurs at a site between the whisker and VPm.

3) The text needs copyediting. A number of sentences could be reworded to clarify the authors' meaning, including:

A) Results: the phrase "Astrocytes solely express IP3R2" could be interpreted to mean that IP3R2 is all that astrocytes express. Instead the authors mean that astrocytes are the only cell (as opposed to microglia and neurons) that express IP3R2.

B) Results: "If this is the case, one direct evidence": wording here could be clarified, such as "… one direct evidence would be the rescue of the refinement defect with application of the P2Y1 agonist".

C) The following sentence in the Discussion should be clarified as: "Due to a close linkage proposed between LTD and synapse elimination, one possible model is that ATP and downstream purinergic signaling could recognize unwanted synapses."

[Editors’ note: a previous version of this study was rejected after peer review, but the authors submitted for reconsideration. The previous decision letter after peer review is shown below.]

Thank you for submitting your work entitled "Astrocytes contribute to synapse elimination via type 2 inositol 1,4,5-trisphosphate receptor-dependent release of ATP" for consideration by *eLife*. Your article has been favorably evaluated by Gary Westbrook (Senior Editor) and two reviewers.

Our decision has been reached after consultation between the reviewers. Based on these discussions and the individual reviews below, we regret to inform you that your work will not be considered further for publication in *eLife*. However, if after considering the reviewer comments, you decide to address the comments with additional experiments or extensive rewriting, we would be willing to consider another submission.

The manuscript was extensively discussed by the reviewers and the editor. As you will see from reading the reviews, there was some difference of opinion at the time the reviews were submitted. However, after discussion both reviewers agreed that the findings that mutations in IP3R2 and P2Y1 lead to defects in developmental connectivity are interesting observations that could be worthy of publication in *eLife*. The work implicating the purinergic system in synapse elimination was also considered interesting, but the data suggesting that ATP acts by inducing presynaptic LTD was less compelling. We agreed that either additional experiments were required or that the manuscript required extensive rewriting to focus on the conclusions that are best supported by the data. Because it is not clear that this could be accomplished within 2 months, the timeframe *eLife* considers the maximum for revision, we are returning the manuscript to you.

*Reviewer #1:*

This manuscript by Yang et al. examines the contribution of IP(3)R2 dependent calcium signaling in astrocytes to developmental synapse elimination at the Pr5-VPm synapse. The authors use a IP(3)R2 knockout mouse (*Itpr2^–/–^*) that exhibits disrupted calcium signaling in astrocytes, combined with manipulations of CSF ATP, P2Y1 receptor agonists and P2Y1 KO mouse, to argue for a model where ATP is release from astrocytes through an activity- and IP3R2-dependent pathway. ATP then activates P2Y1 receptors on presumably presynaptic terminals, inducing a reduction of release probability, leading to elimination of presynaptic inputs.

Although the authors do show that both *Itpr2^–/–^* and P2Y1^–/–^ mice exhibit abnormal convergence of afferent inputs onto relay neurons, their conclusions with regard to the model are overstated as they fail to consider and rule out alternative explanations including,

1) IP(3) R2 has been shown to be present in neuronal processes in the CNS (Holtzclaw et al., Glia 2002). The authors do not convincingly demonstrate that their manipulation targets only astrocytes and not neurons in the circuit, including relay neurons and afferent inputs onto relay neurons.

2) A previous study by one of the authors showed that sensory deprivation leads to defects in refinement at this synapse. In this study, the authors do not rule out the possibility that disruption of IP(3)R2 or P2Y1 (in astrocytes, glia or neurons) might alter the nature of sensory information transmitted from the whiskers to thalamus. To clearly support the authors' hypothesis, cell-type and region specific KOs are needed.

3) The phenotype of *Itpr2^–/–^* mice could be a nonspecific sequela of reduced ATP and global metabolic stress leading to halted or disrupted development.

4) Rescue by ATP in *Itpr2^–/–^* mice may be through an entirely different pathway from astrocyte signaling. For example, injection of ATP could activate microglia-mediated synapse elimination. P2RY1 has been reported to be expressed in microglia (Ballerini et al., 2005 Int J Immunopathol Pharmacol.18(2):255-68) and ATP activates microglia (Davalos et al., Nat Neurosci. 2005 Jun;8(6):752-8).

Specific Comments:

1) Figure 1: More detail is needed on how the authors are counting VGlut2 puncta. For example, in Figure 1 or 2E, how do the authors distinguish between one large puncta from multiple smaller puncta that are so close together that they look like one large object? Are they counting just the puncta that are contacting the NeuN stain? How is puncta/neuron calculated?

2) Figure 1—figure supplement 1: Not clear what region of the brain this is. In addition to somas, there appears to be a lot of processes that are labeled for IP3R2. The low power magnification makes it difficult to assess whether these processes come from astrocytes, glia or neurons.

3) Figure 1—figure supplement 2: How are neurons distinguished from microglia here?

4) According to the authors’ model, PPRatio of the Pr5-VPm synapse in *Itpr2^–/–^* mice should increase, but this was not examined.

5) Figure 3: distribution of *Itpr2^–/–^* + adenosine does not look correct: the sum of the percentages appears to add up to ~55%.

6) Figure 3—figure supplement 1: difficult to visualize the different cell types. Need higher magnification.

7) Figure 5: synaptic recordings show slow and distorted currents, suggesting poor voltage clamp.

8) The authors try to correlate increasing ATP levels in VPm with synapse elimination; however, timepoints during the period of active pruning at this synapse (between P7 and P12, Arsenault and Zhang, 2006) are necessary to support their argument.

9) The authors’ model lacks a mechanism of synapse specificity.

10) Figure 5—figure supplement 1: Images are too low in magnification to rule out the possibility that P2Ry receptors are in the distal processes of microglia.

11) Does intraventricular injection of ATP or adenosine affect synaptic refinement in WT mice?

*Reviewer #1 (Additional data files and statistical comments):*

Higher resolution images that assess the presence of Ip(3)R2 and P2RY1 in cell type specific processes are important supporting data for the authors' model.

*Reviewer #2:*

The overall finding of this paper, that knocking out somatic calcium signaling in astrocytes leads to defective neuronal synapse elimination in the VPm, is novel and interesting. The authors propose that the synapse elimination defect is due to a decrease in astrocytic ATP release in the absence of astrocyte calcium signaling, and that ATP acts via presynaptic P2Y1 receptors to induce LTD at weak synapses, which leads to their elimination. The series of experiments to determine if calcium-dependent astrocyte release of ATP is mediating synapse elimination by inducing LTD via P2Y1 receptors could each have an alternative explanation by themselves, but taken together make a strong case in support of this hypothesis. For example, the P2Y1 KO phenocopies the IP3R2 KO, in terms of synapse elimination. Injection of a P2Y1 agonist to the IP3R2 KO mouse rescues the synapse elimination phenotype. The authors provide an extensive discussion of alternative explanations for their data, along with control experiments that they did to test these alternatives.

The authors have addressed all of the points that I raised when reviewing a previous version of this manuscript, and added additional data figures (Figure 3 and Figure 6).

---

## [Author Response]

*The manuscript is much improved with these changes. There are just a few remaining suggestions to be addressed: 1) It should be made very clear in the figure legend of Figure 1—figure supplement 1 that the co-expression data of IP3R2 was done in the hippocampus, and that the authors could not obtain similar data in VPm because the same antibodies that worked for hippocampus did not work for thalamus.* We thank the reviewers for this suggestion. To clarify, we added a sentence in the Figure 1—figure supplement 1 legend to state that the images were obtained from the hippocampus but not the VPm because the same antibodies (Santa cruz, sc-7278; Millipore, AB3000) that worked for hippocampus did not work for VPm. These changes could be found in the legend for Figure 1—figure supplement 1.

*2) In the third paragraph of the Discussion: The authors argue against the possibility that their synapse elimination findings may arise from altered presynaptic activity because ATP and ATPγS rescue the phenotype. However, they cannot rule out that this rescue occurs at a site between the whisker and VPm.* We agree with the reviewers that the rescue effects by ATP and ATPγS may occur at a site between the whisker and VPm. We re-discuss this issue in the third paragraph of the Discussion.

*3) The text needs copyediting. A number of sentences could be reworded to clarify the authors' meaning, including: A) Results: the phrase "Astrocytes solely express IP3R2" could be interpreted to mean that IP3R2 is all that astrocytes express. Instead the authors mean that astrocytes are the only cell (as opposed to microglia and neurons) that express IP3R2.*

We thank the reviewers for pointing out this confusing statement. Among three subtypes of IP3 receptors, IP3R2 is the only one that expressed in astrocytes. In addition, IP3R2 is not expressed in microglia and neurons in the cerebrum. To clarify, we modified the sentence in the first paragraph of the Results.

*B) Results: "If this is the case, one direct evidence": wording here could be clarified, such as "… one direct evidence would be the rescue of the refinement defect with application of the P2Y1 agonist".*

According to the reviewers’ suggestion, we modified the sentence: “If this is the case, one direct evidence would be the rescue of the synapse elimination defect with application of the P2Y1 agonist in *Itpr2^−/−^* mice”.

*C) The following sentence in the Discussion should be clarified as: "Due to a close linkage proposed between LTD and synapse elimination, one possible model is that ATP and downstream purinergic signaling could recognize unwanted synapses."*

We thank the reviewers for this suggestion and have changed the sentence in the last paragraph of the Results.

[Editors’ note: the author responses to the previous round of peer review follow.]

The manuscript was extensively discussed by the reviewers and the editor. As you will see from reading the reviews, there was some difference of opinion at the time the reviews were submitted. However, after discussion both reviewers agreed that the findings that mutations in IP3R2 and P2Y1 lead to defects in developmental connectivity are interesting observations that could be worthy of publication in eLife. The work implicating the purinergic system in synapse elimination was also considered interesting, but the data suggesting that ATP acts by inducing presynaptic LTD was less compelling. We agreed that either additional experiments were required or that the manuscript required extensive rewriting to focus on the conclusions that are best supported by the data. Because it is not clear that this could be accomplished within 2 months, the timeframe eLife considers the maximum for revision, we are returning the manuscript to you. Reviewer #1: This manuscript by Yang et al.

*examines the contribution of IP(3)R2 dependent calcium signaling in astrocytes to developmental synapse elimination at the Pr5-VPm synapse. The authors use a IP(3)R2 knockout mouse (Itpr2^–/–^) that exhibits disrupted calcium signaling in astrocytes, combined with manipulations of CSF ATP, P2Y1 receptor agonists and P2Y1 KO mouse, to argue for a model where ATP is release from astrocytes through an activity- and IP3R2-dependent pathway. ATP then activates P2Y1 receptors on presumably presynaptic terminals, inducing a reduction of release probability, leading to elimination of presynaptic inputs. Although the authors do show that both Itpr2^–/–^ and P2Y1^–/–^ mice exhibit abnormal convergence of afferent inputs onto relay neurons, their conclusions with regard to the model are overstated as they fail to consider and rule out alternative explanations including, 1) IP(3) R2 has been shown to be present in neuronal processes in the CNS (Holtzclaw et al., Glia 2002). The authors do not convincingly demonstrate that their manipulation targets only astrocytes and not neurons in the circuit, including relay neurons and afferent inputs onto relay neurons.*

We thank the reviewer for raising this important concern. As the reviewer mentioned, Holtzclaw et al. showed that IP3R2 was expressed not only in astrocytes, but also in neuronal processes in the cerebellum. Also, they found that IP3R2 was expressed in somata of few cells with neuronal profile in hippocampus and cortex, which is controversial from other studies suggested that IP3R2 was not expressed in neurons of these brain regions (Hertle and Yeckel, 2007; Li et al., 2015; Petravicz et al., 2014; Sharp et al., 1999). Our immunostaining results in hippocampus are in line with the opinion that IP3R2 was not expressed in the neurons, as showed in Figure 6 and Figure 1—figure supplement 1. Unfortunately, we tried different types of IP3R2 antibodies (Santa cruz, sc-7278; Millipore, RB3000) and found no cells could be labeled in the VPm. In this study, we use IP3R2 deficient mice to selectively eliminate Ca^2+^ signaling in astrocytes but not in neurons, which is further confirmed by calcium imaging experiments as showed in Figure 1—figure supplement 2. Moreover, IP3R2 knockout mice have been extensively used to study the contribution of Ca^2+^ signaling in astrocytes for synaptic transmission (Martin et al., 2015; Petravicz et al., 2008), synaptic plasticity (Agulhon et al., 2010; Di Castro et al., 2011; Navarrete et al., 2012), cerebral blood flow (Bonder and McCarthy, 2014; Nizar et al., 2013), neuroprotection after brain injury (Hertle and Yeckel, 2007) and depression (Cao et al., 2013). These literatures together indicate that IP3R2-KO mice are well accepted in the field as a mouse model for studying the specific role of Ca^2+^ signaling in astrocytes.

Author response image 1.**DOI:**
http://dx.doi.org/10.7554/eLife.15043.017

*2) A previous study by one of the authors showed that sensory deprivation leads to defects in refinement at this synapse. In this study, the authors do not rule out the possibility that disruption of IP(3)R2 or P2Y1 (in astrocytes, glia or neurons) might alter the nature of sensory information transmitted from the whiskers to thalamus. To clearly support the authors' hypothesis, cell-type and region specific KOs are needed.* We agree with the reviewer that cell-type and region specific KOs’ experiments are very important, and needed to be carried out in the future. In the current study, we found that both of RTP and one selective P2Y1 receptors agonist MRS-2365 could rescue the deficient of synapse elimination in IP3R2-KO mice. These results indicate no matter the nature of sensory information has been changed or not in the IP3R2-KO mice, RTP and its downstream purinergic signaling are sufficient to rescue the defects of synapse elimination in the mutants. We discuss this issue in the third paragraph of the Discussion.

*3) The phenotype of Itpr2^–/–^ mice could be a nonspecific sequela of reduced ATP and global metabolic stress leading to halted or disrupted development.* We agree with the reviewer that global metabolic stress caused by reduced RTP may be one of the reasons leading to disruption of synapse elimination in *Itpr2^–/–^*mice, but this is very unlikely. First, these mutant mice have normal brain cytoarchitecture, neuron and astrocyte numbers, and comparable body weight as well as lifespan with WT mice. Second, we found that impaired synapse elimination in *Itpr2^–/–^*mice could be rescued by either RTPçS, a non-hydrolyzable RTP analog, or MRS-2365, a selective P2ry1 agonist. Both molecules could barely provide energy. We discuss this issue in the fourth paragraph of the Discussion.

*4) Rescue by ATP in Itpr2^–/–^ mice may be through an entirely different pathway from astrocyte signaling. For example, injection of ATP could activate microglia-mediated synapse elimination. P2RY1 has been reported to be expressed in microglia (Ballerini et al., 2005 Int J Immunopathol Pharmacol.18(2):255-68) and ATP activates microglia (Davalos et al., Nat Neurosci. 2005 Jun;8(6):752-8).* We agree with the reviewer that rescue effects by RTP may be through activation of microglia. At present, there are mRNR expressions and functional evidences (pharmacological data) showing that microglia express P2Y1 receptors, but the protein evidence is very few. In our study, we did not observe any P2Y1 receptors expression in VPm microglia (Figure 4—figure supplement 1). So, the expression level of P2Y1 receptors in microglia of the VPm, if any, may be too low to be detected by antibody. However, we cannot totally exclude the possibility that RTP activates microglia P2RY1 to rescue synapse elimination defects in IP3R2-KO mice. We re-discuss this in the last paragraph of the Discussion.

Specific Comments:

*1) Figure 1: More detail is needed on how the authors are counting VGlut2 puncta. For example, in Figure 1 or 2E, how do the authors distinguish between one large puncta from multiple smaller puncta that are so close together that they look like one large object? Are they counting just the puncta that are contacting the NeuN stain? How is puncta/neuron calculated?*

We count the number of VGlut2 puncta manually by using the Cell Counter plugin of ImageJ software. In most cases, the VGlut2 puncta are separated from each other. Sometimes there are multiple puncta that stay very close, but we can still discriminate them by the little gaps among them (indicated by the arrow between the No. 2 and 3, No. 4, 5, 6 in Figure 7). Occasionally, there is a big puncta that has no gap, which is then referred as one puncta. We count the puncta that contact the NeuN as *puncta/soma*. The *puncta/neuron* is determined by all the puncta number divided by all neuron number in a given image. We also described this method in detail in the Materials and methods, subsection “Quantification of VGluT2”.

Author response image 2.**DOI:**
http://dx.doi.org/10.7554/eLife.15043.018

Author response image 3.**DOI:**
http://dx.doi.org/10.7554/eLife.15043.019

Author response image 4.**DOI:**
http://dx.doi.org/10.7554/eLife.15043.020

*2) Figure 1—figure supplement 1: Not clear what region of the brain this is. In addition to somas, there appears to be a lot of processes that are labeled for IP3R2. The low power magnification makes it difficult to assess whether these processes come from astrocytes, glia or neurons.*
Figure 1—figure supplement 1 shows IP3R2 immunostaining from hippocampus CR1 region. We repeated this experiment again and found IP3R2 was almost expressed in astrocytes. We also found that microglia and neuronal processes did not express IP3R2, as showed in the new high power magnification Figure 1—figure supplement 1. Unfortunately, although we tried different types of IP3R2 antibodies including (Santa cruz, sc-f2f8; Millipore, RB3000), there were no cells in the VPm could be labeled.

*3) Figure 1—figure supplement 2: How are neurons distinguished from microglia here?* We distinguish neurons from microglia by the big size of cell body.

Neurons normally have bigger cell body than microglia and astrocyte do. In addition, by using CZ3CR1-GFP mice we found that microglia cannot be labeled by calcium dye (Fluo-4 RM and Cal-520).

*4) According to the authors’ model, PPRatio of the Pr5-VPm synapse in Itpr2^–/–^ mice should increase, but this was not examined.* According to the reviewer’s suggestion, we calculated the PPRatio (100ms interval) and found that there was no significant difference between WT and *Itpr2^–/–^* mice. This could be explained by a compensatory mechanism occurred during the embryonic knocking out of IP3R2. In addition, we agree with the editors’ opinion and remove our model of RTP induced LTD causing synapse elimination. We also extensively rewrote the Discussion.

*5) Figure 3: distribution of Itpr2^–/–^ + adenosine does not look correct: the sum of the percentages appears to add up to ~55%.* We thank the reviewer for pointing out this and have corrected the mistake in our revised manuscript.

*6) Figure 3—figure supplement 1: difficult to visualize the different cell types. Need higher magnification.* We thank the reviewer for this suggestion. A higher magnification image is provided in the revised manuscript.

*7) Figure 5: synaptic recordings show slow and distorted currents, suggesting poor voltage clamp.* We replaced the original sample traces with new one in our revised manuscript.

*8) The authors try to correlate increasing ATP levels in VPm with synapse elimination; however, timepoints during the period of active pruning at this synapse (between P7 and P12, Arsenault and Zhang, 2006) are necessary to support their argument.* We measured the RTP levels at P12-P13 in IP3R2-KO mice and found no significant difference when compared to P7-P8. This is not surprising because pruning of VPm relay synapses has different stages. An early stage from P7 to P12 which is insensitive whisker deprivation and a late stage from P12 to P16 which is largely rely on sensory experience. Thus, RTP dependent synapse elimination may only contribute to the late stage of synapse elimination.

*9) The authors’ model lacks a mechanism of synapse specificity.* We agree with the reviewer that our previous model lacks a

mechanism of synapse specificity. According to the editor’s suggestion, we removed our model of RTP induced LTD causing synapse elimination. We also extensively rewrote the Discussion and gave a speculation of how RTP and purinergic signaling recognized unwanted synapses (Discussion, last paragraph).

*10) Figure 5—figure supplement 1: Images are too low in magnification to rule out the possibility that P2Ry receptors are in the distal processes of microglia.* We thank the reviewer for this suggestion. In the revised manuscript, we replace the images with higher magnification ones, as showed in Figure 4—figure supplement 1. We did not observe P2Y1 receptors expressed in the distal processes of microglia.

*11) Does intraventricular injection of ATP or adenosine affect synaptic refinement in WT mice?*

We thank the reviewer for this suggestion. We preformed new experiments with injection of RTP or adenosine in WT mice from P11 to P15 and found no effects on synapse elimination, as showed in Figure 10.

Author response image 5.**DOI:**
http://dx.doi.org/10.7554/eLife.15043.021